# Prevalence of Self-Reported Food Allergies and Their Association with Other Health Conditions among Adults in Saudi Arabia

**DOI:** 10.3390/ijerph18010347

**Published:** 2021-01-05

**Authors:** Nora A. Althumiri, Mada H. Basyouni, Norah AlMousa, Mohammed F. AlJuwaysim, Nasser F. BinDhim, Saleh A. Alqahtani

**Affiliations:** 1Sharik Association for Health Research, Riyadh 13326, Saudi Arabia; mada.basyouni@sharikhealth.net (M.H.B.); nasser.bindhim@sharikhealth.net (N.F.B.); 2Ministry of Health, Riyadh 11176, Saudi Arabia; 3Imam Abdulrahman Bin Faisal University, Dammam 31441, Saudi Arabia; 2170001797@iau.edu.sa; 4King Faisal University, AlAhsa 31982, Saudi Arabia; 216006361@student.kfu.edu.sa; 5College of Medicine, Alfaisal University, Riyadh 11533, Saudi Arabia; 6Saudi Food and Drug Authority, Riyadh 13513, Saudi Arabia; 7Liver Transplant Centre, King Faisal Specialist Hospital & Research Centre, Riyadh 11211, Saudi Arabia; salqaht1@jhmi.edu; 8Division of Gastroenterology and Hepatology, Johns Hopkins University, Baltimore, MD 21218, USA

**Keywords:** Saudi Arabia, food allergy, allergens, prevalence, risk factors

## Abstract

Food allergies are a potentially life-threatening health issue, and few studies have determined their prevalence throughout Saudi Arabia. The main objective of our study was to estimate the prevalence and distribution of self-reported food allergies, and explore their association with other health conditions among adults in Saudi Arabia. This study was a nationwide cross-sectional survey conducted via phone interviews in June 2020. A proportional quota-sampling technique was used to obtain equal distributions of participants by age and gender across the 13 regions of Saudi Arabia. Self-reported food allergy, height, weight, health conditions, mental health status, and demographic variables were collected. Of the 6239 participants contacted, 4709 (75.48%) participants responded and completed the interview. Furthermore, 50.1% of the participants were female, with a mean age of 36.4 ± 13.5 years (18–90 years). The prevalence of food allergies was 19.7%. The most self-reported food allergies were egg, shellfish and shrimp, and peanuts, with a prevalence of 3.7%, 3.1%, and 3.0%, respectively. There was an association between the presence of food allergies and reported bariatric surgery, asthma, colon disease, and being at risk of depression. Bariatric surgery was significantly associated with lower likelihood of self-reported allergy (OR 1.69, 95% CI 1.22–2.34, *p* = 0.002). This study revealed, for the first time, a high prevalence of self-reported food allergies among adults in Saudi Arabia in a large nationwide sample, and food allergy association with bariatric surgery, asthma, colon disease, and being at risk of depression. This information is valuable for clinicians and policymakers, particularly in terms of food allergen labeling.

## 1. Introduction

Food allergies are an “adverse health event arising from a specific immune response that occurs reproducibly on exposure to a given food” [1,2]. Depending on the severity of the food allergy, reactions to foods can range from tingling or itching in the mouth, to severe and possibly life-threatening allergic reactions called anaphylaxis [3]. The severity of food reactions depends on the amount of ingested food, its stability against digestion, and epithelial permeability [4]. Family history, age, asthma, and having other food allergies are risk factors for having allergic reactions to certain types of food [3]. The most common foods which can be eaten separately or included as an ingredient, even in trace amounts, that elicit hypersensitivity reactions are milk, eggs, wheat, fish, and nuts [5]. In general, 90% of food allergies are to milk, peanuts, wheat, fish, tree nuts, shellfish and shrimp, soy, and eggs [6]. However, food allergens and prevalence of food allergies are known to vary regionally [7].

There is convincing evidence that food allergy prevalence is increasing across the world [8]. In fact, research suggests that rates of emergency department visits related to food allergies is increasing among children and young adults [9,10,11]. However, in a survey of 83 World Allergy Organization member countries and six non-member countries, more than half (52 out of 89) had no data on food allergy prevalence [8]. Many of these studies examined the prevalence of food allergies among children or patients, with far fewer focused on the prevalence of food allergies among adults or in community settings [8].

In Saudi Arabia, most studies have been conducted among children, patients, students, or in specific regions [12,13,14,15,16,17]. For instance, a study that was carried out at allergy clinics on 103 patients (79% children) found that prevalence of food allergies was 1.5%, among those 14 years and older and 6% among pediatric patients [12]. Another study of university students found diagnosed food allergies in 9.6% of the 5497 participants [17]. However, they found that there was a lack of public understanding of food allergy diagnoses and management in Saudi Arabia [16]. Thus, poor food labeling and lack of awareness are among the major challenges related to food allergy management [2]. To our knowledge, there have been no nationwide studies investigating the prevalence of food allergies among adults in Saudi Arabia in a community setting. Thus, this study aims to examine the prevalence of self-reported food allergies and specific food allergens among adults in Saudi Arabia, and to explore the association between the presence of food allergies and comorbidities, mental health status, and demographic characteristics.

## 2. Method

### 2.1. Study Design

For this study, we conducted a nationwide cross-sectional survey in Saudi Arabia by phone interviews in June 2020.

### 2.2. Sampling and Sample Size

A proportional quota sampling technique was employed to acquire an equal distribution of participants in terms of age and gender across the 13 regions of Saudi Arabia (Riyadh, Macca, Eastern Province, Asir, Baha, Jazan, Najran, Madinah, Qassim, Hail, Tabuk, Northern Border, and Al-Jouf). Two age groups based on the median age of Saudi Arabian adults (36 years) were used, leading to a quota of 52. The required sample size was calculated based on a medium effect size of approximately 0.25, with 80% power and 95% CI, to compare the age and gender across regions [18]. Thus, each quota required 90 participants, and the total targeted sample size for this study was 4680 participants. The QPlatform^®^ data collection system, which has integrated eligibility and sampling modules, was used to control the sample distribution [19]. The eligibility module included three questions to determine the adherence to the sampling quota, on age, gender, and region.

### 2.3. Participant Recruitment

Participant recruitment was limited to Arabic-speaking adult (≥18 years old) Saudi residents. A random phone number list was generated from the Sharik Association for Health Research to identify potential participants [20]. The Sharik database comprises individuals who are interested in participating in research projects. The database contains more than 70,000 individuals distributed across the 13 regions of Saudi Arabia and continues to grow [20]. Potential participants were contacted by phone on up to three occasions. If there was no response, a new number with similar demographics was generated from the database. After obtaining consent to participate, the interviewer assessed the eligibility, based on the above-mentioned quota completion criteria. Once the quota was complete, it was closed automatically.

### 2.4. Survey and Outcome Measures

Demographic information (age, gender, education level, and region) was captured in the survey. Health conditions were assessed by asking the participants, “Have you ever been diagnosed by a doctor with any of the following chronic conditions?” The conditions included were hypertension, diabetes, hypercholesterolemia, asthma, colon disease, arthritis, sleep apnea, heart disease, cancer, kidney disease, liver disease, thyroid disease, celiac disease, and gastric ulcer. The answer was validated by asking whether the participants were currently taking any medication, or under treatment for the same condition. Food allergies were assessed by asking participants if they were allergic to any of the eight most common food allergens; participants were also able to report allergies to other foods, if applicable. BMI was calculated from self-reported height and weight. BMI was categorized into four groups: underweight (<18.5 kg/m^2^), normal weight (18.5 to 24.9 kg/m^2^), overweight (25 to 29.9 kg/m^2^), and obese (≥30 kg/m^2^). The Patient Health Questionnaire (PHQ-9), a mental health screening tool, was used. The PHQ-9 is well validated and has been used for mental health screening in various national surveys in Saudi Arabia [21,22].

After the first draft of the survey was finalized, a linguistic validation to ensure clarity and understanding of questions was conducted via a focus group, where members were asked to discuss and answer the survey. According to the results of the focus group and feedback from the researchers and interviewers, the questionnaire was further edited, and a final version was produced.

### 2.5. Primary Outcomes of Interest

1.Prevalence of food allergies in the sample of Saudi Arabian adults.2.Associations between self-reported food allergy, demographics, and health-related conditions.

### 2.6. Ethical Considerations

The ethics committee of Sharik Association for Health Research approved this research project (Approval no. 2020-3), in accordance with national research ethics regulations. Participant consent was obtained verbally during the phone interview with the participants and recorded in the data collection system; no call recording took place in this study.

### 2.7. Data Analysis

Descriptive analysis was used to describe the variables. Prevalence results were weighted to equal the adult population in Saudi Arabia, according to the General Authority of Statistics 2017 Census Report [23]. Bivariate analysis (chi-square) was used to compare categorical data. Significant associations were then included in a logistic regression model controlling for age and gender.

## 3. Results

### 3.1. Demographics and Response Rate

Of the 6239 participants contacted, only 4709 (75.48%) participants responded and agreed to complete the interview. There was an equal distribution between the 13 administrative regions of Saudi Arabia. Of the total sample, 50.1% were female, the mean age was 36.4 ± 13.5 years (18–90 years), the median age was 36 years, and 59.6% had a bachelor’s degree or higher. Table 1 shows the demographic characteristics of the participants.

### 3.2. Prevalence of Food Allergy

Overall, the weighted national prevalence of Saudi adults who reported having at least one current food allergy was 19.7%, of whom 16.5% have an allergy to one food and 3.9% have an allergy to more than one food. The proportion of participants in the Najran (*p* < 0.001) and Qassim (*p* < 0.001) regions with food allergies was significantly higher than in other regions based on chi-square ad-hoc analysis. In addition, 17.8% of those who reported having a diagnosed health condition reported at least one food allergy. Table 2 shows distribution and estimation of food allergies among participants.

### 3.3. Association between Food Allergies and Diagnosed Health Conditions and Mental Health

This study shows that there was a significant association between participants reporting food allergies and being diagnosed with asthma (X^2^ (df = 1, N = 4709) = 39.89, *p* < 0.001), colon disease (X^2^ (df = 1, N = 4709) = 23.83, *p* < 0.001), and bariatric surgery (X^2^ (df = 1, N = 4709) = 27.21, *p* < 0.001), in which participants with these health outcomes were more likely to report having food allergies. This study also investigated the association between food allergies and mental health, and found that participants reporting food allergies were more likely to be at risk of depression (X^2^ (df = 1, N = 4709) = 111.01, *p* < 0.001). There was no association between food allergies and the other health conditions assessed.

A logistic regression model including asthma, colon disease, and bariatric surgery controlling for age and gender showed that only bariatric surgery was significantly associated with a lower likelihood of self-reported allergy (OR 1.69, 95% CI 1.22–2.34) *p* = 0.002.

### 3.4. Common Food Allergens

The most common self-reported food allergens were egg (3.7%), shellfish and shrimp (3.1%), and peanut (3.0%), and 8.8% of participants reported allergies to other foods (Table 3).

## 4. Discussion

This was a cross-sectional study that explored the prevalence of self-reported food allergies and specific food allergens among adults in Saudi Arabia, and the association between the presence of food allergies and comorbidities, mental health status, and demographic characteristics through a nationwide phone survey. The present population-weighted data revealed that an estimated 19.7% of adults in Saudi Arabia have at least one type of food allergy. Egg, shellfish and shrimp, and peanuts were the most frequently reported food allergens. Although there were no significant differences between prevalence of self-reported food allergies for gender and age group, there was a significant difference between regions. In addition, there were significant associations between reports of food allergies and presence of asthma, colon disease, bariatric surgery, and risk of depression. Logistic regression analysis revealed a significant association between bariatric surgery and a lower likelihood of self-reported food allergy.

To our knowledge, this is the first study investigating the prevalence of self-reported food allergies among adults in Saudi Arabia at the community level. Findings suggest that the prevalence of self-reported food allergies in this sample was relatively high (21.4%). These results are similar to, or higher than, other studies conducted in the Middle East on the prevalence of self-reported food allergies. The prevalence of self-reported food allergies was 8% among university students in the United Arab Emirates [24] and 12% among university students in Kuwait [25]. Results from the present study are much higher than the findings from other work conducted in Saudi Arabia. Research on a smaller sample size in a clinical setting in Saudi Arabia found that only 1.5% of participants over 14 years of age had been diagnosed with food allergies [12]. Another study conducted among 5497 university students in Saudi Arabia found that 9.6% of participants had been diagnosed with food allergies [17]. However, self-reporting food allergies tend to overestimate actual prevalence [8,26], which may explain some of the discrepancies between previous findings of diagnosed allergies and the present study. For instance, the last published data from the 2015 National Health and Nutrition Examination Survey (NHANES) report an overall prevalence of self-reported food allergies in adults as 13%, while physician-diagnosed food allergy prevalence was only 6.5% [27]. Similarly, another study conducted in the US found that 19% of adult participants (*n* = 40,443) self-reported food allergies; however, this prevalence was reduced to 10.8% when including only convincing reports [9]. Thus, comparisons between the present study and others should be made with caution.

This study demonstrated a significant difference between the prevalence of participants who reported food allergies among the 13 regions in Saudi Arabia, mainly Qassim and Najran. The reasons for the variation between regions are not clear. However, it might be related to social factors such as marriage from the same tribe or cousin marriage; thus, the genetics factor may play a role in the high prevalence of food allergies recorded in these two cities.

The most common food allergens reported in the present study were egg, shellfish and shrimp, and peanuts. This finding conflicts with the results of a small clinical study conducted at the allergy clinics at King Khalid University Hospital in Saudi Arabia, which found that the most common allergens among 280 participants were milk (61.96%), egg whites (59.78%), wheat (45.65%), and peanuts (38.04%), with some participants reporting more than one allergen [28]. Limited data on the most common food allergens in Saudi Arabia prevent us from conducting further comparative analysis. However, the patterns of food allergy across countries are quite different, reflecting the varied diets consumed in different countries [8]. For example, unique food allergens include bird’s nest from swiftlets causing allergic reactions in Singapore and Malaysia, and royal jelly causing allergic reactions in Hong Kong [8].

Our results indicate a significant association between food allergies and asthma. This is an expected finding, given that people who have food allergies are more likely to develop asthma symptoms such as wheezing, coughing, and difficulty breathing [29]. One study has found that the prevalence of asthma in the Arabian Gulf Region (AGR), Kuwait, Saudi Arabia, and Qatar, in particular, is higher than in other Eastern Mediterranean countries [30]. Children who have both asthma and food allergies are more likely to have fatal allergic reactions to food and severe asthma, particularly if the asthma is not well controlled [31]. This may provide an interpretation of our finding of the association between food allergy and asthma. Food allergies can make persistent asthma more likely in young children. According to a study from 2017, between 4% and 8% of children with asthma have a food allergy. However, nearly 50% of children with a food allergy have experienced respiratory symptoms during an allergic reaction, including wheezing and shortness of breath [32]. However, there are few studies that have investigated these relationships in adults.

Our study found a significant association between food allergies and colon disease. However, limited data are available on the relationship between food allergy and colon diseases [33]. It appears that an adverse reaction to food items may be associated with symptom onset or exacerbation [33]. A study conducted on patients with colon diseases in England used food-specific immunoglobulin (Ig) G testing with borderline positive tests [33]. Although protein reactions are rare in colon disease, intolerance to poorly absorbed carbohydrates, such as fructose, lactose, sorbitol, and other fermentable sugars and starches, is related to worsening symptoms in patients with colon diseases [33].

Our results also show a positive association between food allergies and risk of depression, a relationship supported in the literature. The presence of food allergies has been shown to negatively impact quality of life. Psychological distress, which includes anxiety, depression, social isolation, and stress, has been demonstrated in children and adolescents with a food allergy [34]. For instance, one study showed that adults and children who are suffering from food allergies have impaired quality of life and higher levels of stress and anxiety [35]. In addition, a recent Canadian study found that teenagers with food allergies are more likely to have depression, attention-deficit hyperactivity disorder (ADHD), and anxiety [36]. Another study on 1300 Australian adolescents, showed that at 14 years old, nearly one-third of teens with food allergies were self-reporting depression and other behavioral problems. However, when researchers examined the reports from mothers of the same teens, the number of those with depression or other mental health problems increased to 46% [34].

This study found that bariatric surgery is significantly associated with lower likelihood of self-reported food allergy. We searched the literature to establish whether similar results were published about this association via PubMed using the general keywords “bariatric surgery” and “food allergy”: out of only 10 results, one case report in 2017 highlighted the increase in food allergy (birch pollen) severity after bariatric surgery [37]. Another report, in 2020, showed a similar increase in severity of the same type (birch pollen) of allergy in 13 patients [38]. Although there is a lack of knowledge about the validity of such association, digestion and digestibility of carbohydrates and proteins critically affect the risk of food allergy development [39].

The major limitation to this study is that food allergy was self-reported. Self-reporting may lead to overestimating food allergy prevalence by three or four times the actual prevalence [8,26]. This may be due to participants’ confusing a food allergy with other adverse reactions to food [8,26]. Data on the public awareness of the difference between food allergies and intolerances are scarce. Accurate reports on the prevalence of food allergies require tests to confirm food allergies, such as skin allergen testing, food item specific allergen serum Immunoglobulin E (IgE), or oral food challenges. Future research efforts could focus on diagnosing food allergies using oral food challenges. On the other hand, underestimation of the number of diagnosed food allergies can also occur, because some people might not seek medical help if they experience mild food allergies or are not aware of their condition due to inadequate exposure to the type of food in question. Furthermore, this study was cross-sectional, precluding conclusions about causation. However, this study design was strengthened by the use of quota sampling, potentially limiting the risk of selection bias and allowing for the recruitment of a balanced study sample in terms of gender and age. A large study sample with high coverage of adults in Saudi Arabia is a strength of this work, especially considering the limited information available on food allergy prevalence in this population. Furthermore, the use of a research participant database might introduce bias, given that participation in the database was voluntary. Nevertheless, such a database is useful for conducting research with large sample sizes without encountering prohibitive barriers, especially considering that the survey took place during the COVID-19 pandemic, which could have otherwise inhibited recruitment strategies. Data integrity checks, inherent to the QPlatform data collection system, minimized invalid or erroneous data entry. Linguistic validation and questionnaire piloting were employed to strengthen the questionnaire’s reliability.

## 5. Conclusions

Prior to this study, most food allergy research in the Saudi Arabian population was collected from specific and non-generalizable settings or regions, or among children. There had not been a nationwide study on Saudi Arabian adults. This study provides insights into the prevalence of self-reported food allergies and sources of allergens among adults. This information is valuable for clinicians and policymakers, particularly in terms of food allergen labeling. This study showed that there was a significant association between participants reporting food allergies and participants with asthma, colon disease, and bariatric surgery. This association may need to be investigated further and may be used as an indicator for the risk of having a food allergy. This study’s results will also open the door for future research in food allergy prevalence and characteristics in Saudi Arabia, and further elaboration is needed to properly distinguish, diagnose, and manage food allergies.

## Figures and Tables

**Table 1 ijerph-18-00347-t001:** Demographic characteristics of participants (*n* = 4709).

Demographic Characteristics	*n* (%)
**Age Groups**	
18–19	255 (5.4)
20–29	1556 (33.0)
30–39	1009 (21.4)
40–49	1044 (22.2)
50–59	555 (11.8)
60+	290 (6.2)
**Gender**	
Female	2358 (50.1)
Male	2351 (49.9)
**Education Level**	
Less than bachelor’s	2370 (50.3)
Bachelor’s and above	2338 (49.6)

**Table 2 ijerph-18-00347-t002:** Prevalence of self-reported food allergies, overall, and by gender, age group, region, and presence of comorbidity in the study sample (*n* = 4709).

Variable	*n* (%)	Chi-Square *p*-Value
Overall	1009 (21.4)	-
**Gender**		
Male	490 (20.8)	0.329
Female	519 (22.0)	
**Age Group**		
18–19	59 (23.1)	0.416
20–29	335 (21.5)	
30–39	228 (22.6)	
40–49	202 (19.3)	
50–59	127 (22.9)	
60+	58 (20.0)	
**Region**		
Al-Jouf	54 (15.0)	<0.001
Northern Border	68 (18.8)	
Tabuk	84 (23.2)	
Hail	81 (22.6)	
Madinah	68 (23.5)	
Qassim	102 (28.3)	
Macca	70 (19.3)	
Riyadh	60 (16.5)	
Eastern Province	70 (19.4)	
Baha	75 (20.6)	
Asir	73 (19.9)	
Jazan	72 (19.9)	
Najran	114 (31.5)	
**Presence of comorbidities**		
No diagnosed health condition	551 (25.8)	<0.001
≥1 diagnosed health condition	458 (17.8)	

**Table 3 ijerph-18-00347-t003:** Prevalence of self-reported specific food allergies in general adult population weighted sample (*n* = 4709).

Type of Food Allergy	*n* (%) ^a^
Egg	172 (3.7)
Shellfish and Shrimp	145 (3.1)
Peanut	141 (3.0)
Milk	123 (2.6)
Fish	118 (2.5)
Tree Nuts	82 (1.7)
Soy	44 (0.9)
Wheat	39 (0.8)
Other	410 (8.8)

**^a^** Participants were able to report food allergy to more than one allergen.

## Data Availability

Available from Sharik Association for Health Research upon request.

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
