# Peer review of "Prevalence of Self-Reported Food Allergies and Their Association with Other Health Conditions among Adults in Saudi Arabia"

_ijerph, 2021, doi:10.3390/ijerph18010347_

Round 1

Reviewer 1 Report

The entire manuscript must be formatted according to the journal’s instructions. For example, authors, affiliations and references are not in accordance with the instructions to authors.

I may advise a thorough review of the entire text before making a new submission. I can give as an example the redundancy in line 29: "The participants were 50.1% were female ..."

More data should be provided in the Introduction. I suggest the inclusion of these references:

Alharbi, R., Malibary, H., Siddiqui, J., & Alandijani, S. (2020). Awareness about diagnosis and management of food allergy among general population in Jeddah, Saudi Arabia. Medical Science24(102), 905-911.

Raposo, A., Pérez, E., de Faria, C. T., & Carrascosa, C. (2017). Allergen management as a key issue in food safety. Food Safety and Protection.

In the Materials and Methods section should be specified the 13 Saudi Arabian regions included in the study.

The discussion section of the present study should be deepened taking into account similar studies carried out in other countries, providing more details in the comparison between the different studies and considering the specific characteristics of the culture and geographic location where these same studies were developed.

Bearing in mind the pertinence of the theme and the work developed, I think that with the necessary modifications this work can be published on IJERPH.

Author Response

Re: [ ijerph-1051969] “Prevalence of self-reported food allergies and its association with
other health conditions among adults in Saudi Arabia”

Thank you for your valuable comments and suggestion which we believe helped in improving the manuscript.

Reviewer #1:

1- The entire manuscript must be formatted according to the journal’s instructions. For example, authors, affiliations and references are not in accordance with the instructions to authors.

Authors’ Response: Agree, all text been review by the authors and MDPI English Editing Services.   

2- I may advise a thorough review of the entire text before making a new submission. I can give as an example the redundancy in line 29: "The participants were 50.1% were female ..."

Authors’ Response:  Agree, all text been review by the authors and MDPI English Editing Services.        

3- More data should be provided in the Introduction. I suggest the inclusion of these references:

Alharbi, R., Malibary, H., Siddiqui, J., & Alandijani, S. (2020). Awareness about diagnosis and management of food allergy among general population in Jeddah, Saudi Arabia. Medical Science, 24(102), 905-911.

Raposo, A., Pérez, E., de Faria, C. T., & Carrascosa, C. (2017). Allergen management as a key issue in food safety. Food Safety and Protection..

Authors’ Response: Done, introduction have been updated.   

4- In the Materials and Methods section should be specified the 13 Saudi Arabian regions included in the study.

Authors’ Response: Done, regions were added.

5- The discussion section of the present study should be deepened taking into account similar studies carried out in other countries, providing more details in the comparison between the different studies, and considering the specific characteristics of the culture and geographic location where these same studies were developed.

Authors’ Response: Agree, we included more information about the role of unique food patterns in common food allergy between countries.

6- Bearing in mind the pertinence of the theme and the work developed, I think that with the necessary modifications this work can be published on IJERPH.

Authors’ Response: Noted with many thanks for reviewing this work.

Reviewer 2 Report

The main objective of this article entitled "Prevalence of self-reported food allergies and its association with other health conditions among adults in Saudi Arabia" is to estimate the prevalence and distribution of food allergies and explore its association with other health conditions among adults Arabia.
The article is about an interesting topic but has important limitations that are detailed below:
1. In the abstract, the conclusions do not mention the association of food allergies with other health conditions. The same happens in the conclusions in the text.
2. Regarding the methodology, the study has gone through an ethics committee as a plus. Have the telephone interviews been recorded? Is there a record of those calls?
3. The manuscript presents important errors in citations in the text, bibliographic references and the structure of the tables. It does not comply with the regulations of the magazine.

Author Response

Re: [ ijerph-1051969] “Prevalence of self-reported food allergies and its association with
other health conditions among adults in Saudi Arabia”

Thank you for your valuable comments and suggestion which we believe helped in improving the manuscript.

Reviewer #2:

The main objective of this article entitled "Prevalence of self-reported food allergies and its association with other health conditions among adults in Saudi Arabia" is to estimate the prevalence and distribution of food allergies and explore its association with other health conditions among adults Arabia.The article is about an interesting topic but has important limitations that are detailed below:

Authors’ Response: Noted with many thanks for reviewing this work.

  1. In the abstract, the conclusions do not mention the association of food allergies with other health conditions. The same happens in the conclusions in the text.

Authors’ Response: Done. The association of food allergies and health conditions was added to both abstract and conclusions.

  1. Regarding the methodology, the study has gone through an ethics committee as a plus. Have the telephone interviews been recorded? Is there a record of those calls?

Authors’ Response: This study gone through an ethics committee that links and approved by national ethics unite King Abdulaziz City of Science and Technology. No calls were recorder. We only take verbal consent from the participants before preforming the interview.  This part has been included in the manuscript for clarification.

  1. The manuscript presents important errors in citations in the text, bibliographic references and the structure of the tables. It does not comply with the regulations of the magazine.

Authors’ Response: Agree, all text been review by the authors and MDPI English Editing Services.

Round 2

Reviewer 1 Report

The manuscript can now be published in IJERPH.

Reviewer 2 Report

I agree with the modifications made by the authors.